# Multi-Perspective Feature Extraction and Fusion Based on Deep Latent Space for Diagnosis of Alzheimer’s Diseases

**DOI:** 10.3390/brainsci12101348

**Published:** 2022-10-05

**Authors:** Libin Gao, Zhongyi Hu, Rui Li, Xingjin Lu, Zuoyong Li, Xiabin Zhang, Shiwei Xu

**Affiliations:** 1College of Computer Science and Aritificial Intelligence, Wenzhou University, Wenzhou 325035, China; 2Key Laboratory of Intelligent Image Processing and Analysis, Wenzhou 325035, China; 3College of Computer and Control Engineering, Minjiang University, Fuzhou 350108, China; 4School of Artificial Intelligence, Wenzhou Polytechnic, Wenzhou 325000, China

**Keywords:** latent space representation, brain disease, long short-term memory, convolutional neural networks, Alzheimer’s disease

## Abstract

Resting-state functional magnetic resonance imaging (rs-fMRI) has been used to construct functional connectivity (FC) in the brain for the diagnosis and analysis of brain disease. Current studies typically use the Pearson correlation coefficient to construct dynamic FC (dFC) networks, and then use this as a network metric to obtain the necessary features for brain disease diagnosis and analysis. This simple observational approach makes it difficult to extract potential high-level FC features from the representations, and also ignores the rich information on spatial and temporal variability in FC. In this paper, we construct the Latent Space Representation Network (LSRNet) and use two stages to train the network. In the first stage, an autoencoder is used to extract potential high-level features and inner connections in the dFC representations. In the second stage, high-level features are extracted using two perspective feature parses. Long Short-Term Memory (LSTM) networks are used to extract spatial and temporal features from the local perspective. Convolutional neural networks extract global high-level features from the global perspective. Finally, the fusion of spatial and temporal features with global high-level features is used to diagnose brain disease. In this paper, the proposed method is applied to the ANDI rs-fMRI dataset, and the classification accuracy reaches 84.6% for NC/eMCI, 95.1% for NC/AD, 80.6% for eMCI/lMCI, 84.2% for lMCI/AD and 57.3% for NC/eMCI/lMCI/AD. The experimental results show that the method has a good classification performance and provides a new approach to the diagnosis of other brain diseases.

## 1. Introduction

Alzheimer’s disease (AD) is a brain disorder that affects memory, thinking and behavior [1,2]. Early stages of AD presentation are often characterized by mild memory loss and reduced learning ability. Symptoms are often insidious, and the disease causes slow deterioration. Identifying the early stages of AD, known as Mild Cognitive Impairment (MCI), remains challenging [3,4]. Early MCI (eMCI), with slight impairments in memory and learning, usually does not affect the basics of daily life; therefore, opportunities for the early treatment of the disease are often missed [5,6]. The present treatments still do not provide a complete cure for AD, but treatment can temporarily slow down the worsening of symptoms, which may still progress and lead to a gradual loss of self-care. Although AD is incurable, accurate diagnosis and intervention in the early stages of AD development, i.e., at the eMCI stage, can still be of great importance in reducing or delaying the progression of the disease.

Resting-state functional magnetic resonance imaging (rs-fMRI) is now widely used in basic research and disease diagnosis [7,8,9,10]. The blood-oxygenation-level-dependent (BOLD) imaging of brain regions can be obtained non-invasively and without radiation using fMRI. The functional connectivity between brain regions is constructed by measuring the temporal correlation of the BOLD signal. The Pearson correlation coefficient (PCC) is mostly used to achieve this, describing the connections between brain regions [11,12]. FC has been widely used in the diagnosis and analysis of brain diseases, enabling a better understanding of the interactions between brain regions and the link between diseases [13,14]. In previous work, FC was assumed to be static during the fMRI process [15,16]. It has been shown that time series with different dynamics have different topologies and construct different FCs [17,18]; thus, using static FC (sFC) would cause the dynamic changes in brain regions to be lost. To capture the time-varying information on the brain, a sliding window-based approach is proposed to construct the dFC, which is used to represent the dynamics of brain regions. Recently, it has been shown that learning the dFC can be of great help in disease analysis and understanding [19,20]. Many studies have not explored the rich spatial and temporal information in the dFC. Hence, the spatial and temporal features of the dFC can be explored in more depth for the better classification of brain disease.

In previous research, PCC was typically used to construct the dFC, after employing some simple metrics to obtain low-level representation for use as features for brain disease classification and analysis [21,22]. However, there are several key limitations to this approach. (1) Using a simple metric to obtain a representation of the brain is often low-level and does not filter out the redundant information that may be present in the representation. Studies have shown that the human brain may consistently use rich contextual high-level information in brain regions to perform high-level tasks, and that it is difficult to describe complex changes in brain regions using a simple metric [23]. (2) Directly using the representational information of the dFC may not be effective in the extraction of globally contextual relevant information for brain regions. (3) The analysis of complex higher-order feature information from a single perspective may overlook local spatial and temporal dynamics and global variability in the features. To address the above problems, this paper proposes a new LSRNet model. First, we propose a feature extraction method based on an autoencoder to learn the latent space representation of dFC and explore the contextual information and latent high-level features in the brain regions in more depth. This enables the latent contextual relevance in dFC to be learned, while filtering commonalities in the representation and extracting high-level latent space representation. Second, the dFC may respond to global contextually relevant features and local features of the spatial and temporal dynamics of brain regions, collaborating on high-level tasks when the brain is performing such tasks. Two models are used to target the global contextually relevant features and the local spatial and temporal dynamics. We propose a global perspective-based convolutional neural network for the parsing and deeper extraction of latent feature representations from a global perspective for global contextually relevant features. For local spatial and temporal dynamic variation features, we use a bidirectional LSTM (BiLSTM) for dynamic feature parsing of the latent feature representation from two directions of the time dimension to obtain high-level features of feature variation. Finally, we fuse the features extracted from the two models in depth, and conduct an assisted diagnosis study of brain disease. We test LSRNet on a constructed test set, and our overall experimental comparison of the models, and comparisons with the results of other research work, verify the reliability of the model and significantly improve the diagnosis of brain disease.

## 2. Materials

In our study, we used the rs-fMRI data of 174 subjects, the ADNI dataset [24,25], including 48 Normal Control (NC), 50 patients with early MCI (eMCI), 45 patients with late MCI (lMCI), and 31 patients with AD. Each subject participated in at least one, and sometimes multiple, scans at the time of the visit, so that these 174 subjects had a total of 563 scans. These 563 scans can be divided into 154 NC cases, 165 eMCI cases, 145 lMCI cases and 99 AD cases. Each rs-fMRI scan was acquired at different medical centers using a 3.0T Philips scanner. For each scan, the flip angle was 80.0 degrees, the image resolution was 2.29–3.31 mm, the slice thickness was 3.31 mm, the TE was 30 ms, the TR was 2.2–3.1 s, the imaging matrix was 64 × 64 pixels, the FoV was 256 × 256 mm${}^{2}$, and the scan time per subject was 7 min (a total of 140 volumes).

We adopted data pre-processing methods that were referenced and consistent with those of several refereed papers [13,21,26]. We performed standard processing procedures on all subjects scanned via rs-fMRI using FSL FEAT. Specifically, we dropped the previous three volumes before preprocessing, which consisted of slice timing correction, head motion estimation, bandpass filtering, and regression of nuisance covariates, with subjects whose head motion was greater than 2 mm rotation or two degrees maximum rotation being excluded. We followed this with structural skull-stripping based on the subject’s corresponding T1-weighted sMRI images; the flip angle was 8–9 degrees, the imaging matrix was 256 × 256 × (160–170) pixels, the FoV was (256–260) × 240 mm${}^{2}$. The stripped data were aligned with the Montreal Neurological Institute space and the images were smoothed using a 6 mm Gaussian filter. Finally, we divided the brain space of the rs-fMRI scans into 116 regions of interest (ROIs) according to the Automated Anatomical Labeling template and extracted the time series of BOLD signals as the initial input data.

We conducted multiple sets of experiments, including multiple 2-category tasks (e.g., NC/eMCI, NC/AD, eMCI/lMCI, lMCI/AD) and a 4-category task (NC/eMCI/lMCI/AD) using a 5-fold cross-validation. Additionally, this paper refers to the approach in the paper [21], where the dataset partitioning is strictly subject-independent, dividing the 174 subjects into 147 subjects who had a baseline scan and 27 subjects who did not have a baseline scan. The 147 subjects were roughly divided into five parts, with four of them being used. The subjects without a baseline scan were used as the training set and the remaining subjects were used as the testing set. In addition, we used 15% training subjects as the validation data to tune the parameters. This ensures that multiple scans of the same patient will be divided into the training or test set, that there is no data leakage, and that data stability is ensured.

For first stage of the experiment, we randomly used the dFCs of the subjects in the training set as input to the autoencoder to enhance the generalization ability of the model and allow for the model to capture the high-level features in dFCs. We input M dFCs of the same subject into the autoencoder to obtain a latent space representation of size M*H as input to the stage 2 model, which will ensure that the model at the stage 2 can understand and learn the spatial and temporal variations and global variations of the same patient.

## 3. Methods

### 3.1. Latent Space Representation Network

As shown in Figure 1 and Figure 2, LSRNet has two learning stages. Stage 1 involves deep latent space representation learning and stage 2 involves multi-perspective latent space representation learning and fusion. The first stage consists of: (1) rs-fMRI preprocessing, (2) construction of the dFC, (3) dFC learning using a deep autoencoder to obtain the latent space representation. Three parts are involved in this stage: (a) a convolutional neural network for the global perspective, (b) a BiLSTM network for the local perspective, (c) a multi-perspective feature fusion network. In this section, the specific structures mentioned above are described in detail.

#### 3.1.1. Stage 1: Deep Latent Space Representation Learning

rs-fMRI pre-processing: The first task is to normalize the ROIs for each subject, and for a BOLD time series t with N ROIs. We obtained the following Equation (Equation 1):
(1)f(t)=(t−μ)/σμ, σ are the mean and variance of the corresponding series, respectively.Construction of the dFC: To describe the dynamic changes in brain regions and construct the dFC, we divided the time series of ROIs obtained by rs-fMRI into *M* overlapping sliding windows. The sliding window size and step size in this paper are based on previous work [13,21,23,26], with a window size of 30 timepoints (90 volumes for a total of 270 s) and a step size of two steps (6 s) to construct M overlapping subsequences $S=\{S_{1},S_{2},\ldots ,S_{M}\}$; for any i=1,2,…,M, we have $S_{i}=\left\{t_{2i-1},t_{2i-1+1},\ldots ,t_{2i-1+29}\right\}$, and *M* is 54 in this paper. According to the PCC, for each overlapping subsequence of *N* ROIs, *M* dFCs are constructed, and we have Equation (Equation 2), as follows:
(2)$$\mathrm{PCC}\left(t^{i},t^{j}\right)=\frac{E\left[\left(t^{i}-\mu_{t^{i}}\right)\left(t^{j}-\mu_{t^{j}}\right)\right]}{\sigma_{t}\sigma_{t^{j}}}$$In the above Equation (Equation 2), $t^{i}$,$t^{j}$; then, denote the i-th and j-th ROI segments in each overlapping subsequence, and $\mu_{t^{i}}$,$\mu_{t^{j}}$ and $\sigma_{t^{i}}$,$\sigma_{t^{j}}$ correspond to the mean and variance of the i-th and j-th segments of that ROI sequence, respectively.Learning of the dFC using a deep autoencoder to obtain the latent space representation: We constructed M dFCs, and since the dFC obtained by PCC is a diagonal matrix, the redundant feature information is removed, the upper triangular part of each diagonal matrix is taken out, and the dimensionality of the features is reduced from $N^{2}$ to N*(N−1)/2. For each subject, we have M*N*(N−1)/2 features, and these are used as the input and label of the autoencoder in stage 1.

We designed the autoencoder for stage 1 as in Figure 1, which takes the input parameter *X* and inputs it to the encoder layer. This performs a multi-layer feature dimensionality reduction on the input parameter to obtain a latent space representation of length H. This is input to the decoder layer, which performs the corresponding multi-layer feature dimensioning of the latent space representation and outputs $\hat{X}$. The feature sizes of *X*, $\hat{X}$ are both N*(N−1)/2. The loss function is constructed by applying the following Equation (Equation 3) to *X*,$\hat{X}$.
(3)$$\mathrm{loss}\left(X,\hat{X}\right)=\sqrt[{n}]{\sum_{i=1}^{n}{\left(X_{i}-{\hat{X}}_{i}\right)}^{2}}$$

Equation (Equation 3) is used to extract the latent feature representation in the dFC by learning the feature differences in *X*,$\hat{X}$. Finally, via the autoencoder, we were able to obtain the latent space representation in the dFC. M latent space representations of the same subject were used as one input to stage 2.

#### 3.1.2. Stage 2: Multi-Perspective Latent Space Representation Learning and Fusion

To capture possible high-level variance features in the latent space representation, we used the methods in (a) and (b) to learn features from different perspectives on the latent space representation.

Convolutional neural networks for the global perspective (Global CNN, GCNN): In order to parse the global contextual information latent in brain regions, we designed a global-perspective-based convolutional neural network to extract global contextual information in the latent space representation. Specifically, for each subject, we set the size of the kernel to M*H. That is, we globally parsed the dynamic features of the brain regions at each moment in time from a global perspective, which can also be interpreted as extracting contextual information as the brain completes high-level tasks, enabling the understanding of multiple complex brain-context-related activity information through multiple convolutional kernels. However, since the latent space representation belongs to the high-level feature representation in the dFC, a convolutional neural network employing a global perspective at this stage could discover global contextual information from the high-level features. The multi-kernel convolutioncan be thought of as understanding the brain’s high-level tasks and extracting the contextual feature information needed to perform these high-level tasks.BiLSTM for the local perspective: In order to parse the local spatial and temporal dynamics of brain regions, we used a BiLSTM to parse the temporal dimension, to parse the features in both directions from the temporal dimension, to obtain the spatial feature variations, and to gradually filter the features at multiple levels from the spatial dimension to finally obtain dynamic high-level information at the spatial and temporal levels. At this stage, it can be understood that the BiLSTM network gradually understands the high-level information of the brain according to the temporal variation, and then abstractly understands the relationship between the dynamic variations in brain regions and disease in different patients.Multi-perspective feature fusion network (FusionNet): We combined the global contextual feature information (global features, GF) of the high-level task with the spatial and temporal dynamic variation features (dynamic features, DF), and compressed and downscaled the two models to perform multi-level deep feature fusion. As shown in Equation (Equation 4), we stacked the feature information from GF and TF and fed this into FusionNet for feature dimensionality reduction and compression, and finally used the softmax layer for feature classification.
(4)diseasepredict=FusionNet(concat(GF,DF))

Finally, we combined information on the understanding of high-level tasks in brain regions with information about dynamic variations in diseases specific to brain regions, which completed the disease diagnosis.

## 4. Results

Our proposed method was implemented using Python and the PyTorch framework, and the model was trained on an NVIDIA Tesla V100 with 32 GB video memory. In the first stage, our autoencoder implementation used a fully connected layer to encode and decode features of N*(N−1)/2 feature dimensions. Each encoder and decoder layer had sizes of 2048, 1024, and 512. To guarantee the model output, we considered the dFC values constructed by the PCC to be between −1 and 1. Fully connected layers in the autoencoder were followed by a normalized and tanh activation function. In the second stage, our global convolutional neural network consists of multiple convolutional kernels of length size M*H and a BiLSTM with a hidden layer size of 32, which is an empirical rule derived from the literature [23]. Finally, we conduction an in-depth fusion of the obtained global high-level features and the local variation features to reduce the dimensionality and, after the reduction, the predicted results were output via softmax.

### 4.1. Methods for Comparison

In this experiment, our proposed method was compared with six methods: two traditional methods (ROIs + SVM, and sFC + SVM) and four variants of the proposed method (dFC + BiLSTM, dFC + GCNN, LSRNet-G, and LSRNet-B).

ROIs + SVM: A simple method for processing BOLD signals, which directly stacks full-length BOLD signals (N*N) from N brain regions, trains the intensity of BOLD as an SVM and predicts brain disease.sFC + SVM: Most traditional research methods involve the construction of sFCs from full-length BOLD information based on the PCC, stacking the sFCs in a reduced dimension and feeding them to SVMs for training.dFC + BiLSTM: In recent studies, the dFC is usually constructed according to a set sliding window, and the upper triangular features of the dFC symmetrical matrix are extracted and input to the BiLSTM model. The BiLSTM learns relevant information about brain regions in both directions and the association between diseases and brain region FC, and finally achieves brain disease diagnosis. Specifically, the upper triangular features are stacked into N*(N−1)/2, the hidden layer features of the BiLSTM are set to *H*, and M*N*(N−1)/2 features of the same subject are used as the model input.dFC + GCNN: As a component of our study, the direct use of GCNN can extract high-level features in the global context of brain regions in the dFC. Specifically, we use *M* dFCs from each subject as input to the global convolutional layer, and the output features are classified via softmax.dFC + AE + GCNN (LSRNet-G): This method removes the BiLSTM compared to the proposed method to evaluate the BiLSTM’s contribution to the model. Specifically, we use the compressed global high-level contextual features for brain disease classification after learning the global perspective of the GCNN’s representation of the latent space of the autoencoder.dFC + AE + BiLSTM (LSRNet-B): This method removes the GCNN and directly uses the BiLSTM to extract the latent space representation for brain region classification. The M output latent space representations are fed into the BiLSTM one at a time, according to the temporal order for extracting dynamic variations in brain regions. Finally, we perform feature stacking and dimensionality reductions on the last layer of the BiLSTM output for brain disease diagnosis classification.

The performance of the full model is summarized in Table 1, Table 2 and Table 3. We can see that our proposed method outperforms all the methods used for comparison in several binary classification tasks and four classification tasks. For the traditional method dFC + BiLSTM, the variant dFC + AE + BiLSTM achieves 80% and 76.3% accuracy in the NC/eMCI classification task and NC/AD achieves an improvement of about 17% and 21%. The GCNN-based variant dFC + AE + GCNN also improved by about 2–10% in the binary classification metric. This demonstrates the effectiveness of using a latent space representation for disease diagnosis. The variant GCNN showed an overall improvement compared to the normal GCNN approach with dFC direct learning. Our variant GCNN performance increased by about 2–10% compared to the normal GCNN in the full binary range and four classification tasks. The experimental results validate that the use of latent space representation can extract high-level contextual features in brain regions, which can enhance and stabilize the model accuracy. Compared with the variant GCNN, the performance of our proposed variant BiLSTM is improved even further. The traditional BiLSTM is unable to effectively filter invalid information from the dFC and extract temporal change features due to the relatively large input features and the lack of feature filtering. The method based on deep latent space representation can effectively filter the commonalities in the dFC and extract dynamic changes in relation to disease. The experimental results validate the effectiveness of the method, and our variant BiLSTM shows an overall increase in performance of about 3% 24% in all aspects compared to the traditional BiLSTM. Our proposed method combines the global perspective of GCNN and the local features of BiLSTM, and the accuracy of the proposed method is more stable. For example, when diagnosing NC/eMCI, the deviations of $ACC_{NC}$ and $ACC_{eMCI}$ of our proposed variant GCNN and variant BiLSTM are larger. The $ACC_{NC}$ of the variant GCNN is 68.8%, while the $ACC_{eMCI}$ is 83.3%, a high level of variation compared to the overall ACC, and the same is true for the variant BiLSTM.

### 4.2. Feature Analysis

In addition, we studied the variations in our proposed latent space representation. Specifically, we extracted latent space representations for the four categories of subjects from the autoencoder, which were learned in the first cross-validation in the four-category task. As shown in Figure 3, a feature matrix consisting of M windows and a latent space size of 512 was drawn. For simplicity, we calculated the average feature matrix for each category. The rows correspond to the feature values of the windows.

From Figure 3, we can obtain some remarkable information. First, the latent space representation obtained by the autoencoder has obvious differences between categories, which indicates that learning by dFC can be more discriminative. Secondly, data from different categories with the same change trends at the same location also significantly different between parts of the same location, reflecting the commonality of changes in brain regions and differences in brain diseases. Finally, we found that the latent space representation is decreasing in some locations as the disease continues to worsen, reflecting the disease’s impact on brain regions. The variability of these characteristics enables us to show the validity of our proposed method.

### 4.3. Comparison with Other Methods

We also compare our method with some of the more recent methods. Since NC/eMCI and NC/AD are more common in mainstream methods, they are used here as metrics for comparison. The results are shown in Table 4, where the method proposed in this paper has the best performance among the many methods. It is worth noting that the dataset in this paper is close to that of the other methods, but there may be some differences. In NC/eMCI, the performance of this paper is close to that of the other methods and, in NC/AD, the method proposed in this paper performs well, with a 1–7% improvement compared to the other methods. The experimental results validate the effectiveness of our model to some extent. It is worth noting that, as the datasets and data pre-processing techniques used in each method are not the same, a direct comparison of the results is not reliable and can only serve as a degree of reference.

## 5. Discussion

At present, dFC is widely used in the diagnosis of brain diseases [13,21,23,26]. The usual approach to constructing the dFC is to use the PCC method to measure the correlation between different brain regions. The main drawback of the PCC approach is that it treats different points in time equally, thus ignoring the specific contributions of different points in time [21]. In this paper, we define the new autoencoder method to measure correlations in brain regions and extract high-level features. The latent space representation of the same subject is extracted by the model to represent the important features of the complex brain regions of that subject. Then, we analyzed the features in depth from global and local perspectives, and finally fused the two forms of feature information through FusionNet to finalize the disease diagnosis. Our proposed method is the first attempt to abstractly represent important changes in brain regions using models and to diagnose disease by two perspectives on the feature changes. The final experimental results show that our proposed method is superior to previously proposed methods, and that our proposed method provides a new perspective to analyze the complex changes in the brain and provides new ideas for other researchers to analyze the relevant features of brain regions.

In previous work, a study was conducted directly by diagnostic analysis of the dFC features of the upper triangle using BiLSTM [23,29], and good results were obtained from this. In our experiments, considering the dynamic variability of brain regions, we also used BiLSTM to study the dynamic variability of latent space representation. In addition, there is a certain degree of variation in the dFC constructed from overlapping windows, and this variation is difficult to capture using the BiLSTM of the local perspective. Inspired by previous studies [21], the window was parsed from a global perspective using GCNN, and the features of both methods were fused and used to diagnose brain diseases.

### Limitations

Although our method achieved a good performance in terms of disease diagnosis, the study is limited by the following three factors. First, only rs-fMRI data were used in this study for the diagnosis of brain diseases. In fact, data from other modalities (e.g., sMRI, clinical data, and genetic data) can also provide complementary information for brain disease diagnosis from different perspectives. In future work, we will integrate data from multiple modalities to study brain diseases. Second, our experiments were validated on subjects with different scan parameters for the proposed model, which may affect the performance of our proposed method. In future work, we will evaluate our model on a dataset with the same scan parameters. Third, this study only used the ADNI dataset, which has a small sample. In future work, we will evaluate the proposed method on more brain diseases, such as autism spectrum disorders.

## 6. Conclusions

In this paper, we propose the use of the LSRNet model for multi-perspective feature learning and fusion based on latent space representation. This is based on the dFC for high-level latent space representation learning, learning the potential contextual relevance in the dFC, and extracting the abstraction of the latent high-level latent space representation. For the dynamic variability of local brain regions and high-level tasks based on contextual information, a spatial and temporal feature capture model of local dynamic variability, and a global contextual feature learning model, are proposed. The two high-level features are deeply fused and finally used for brain disease diagnosis. The experimental results validate the effectiveness of the model in this paper, achieving the best performance in several binary metrics and four classifications. Our approach not only improves the classification performance, but also provides new analytical ideas for other researchers. In the future, a combination of fuzzy systems [30] and the method proposed in this paper will be applied to more brain disease classifications.

## Figures and Tables

**Figure 1 brainsci-12-01348-f001:**
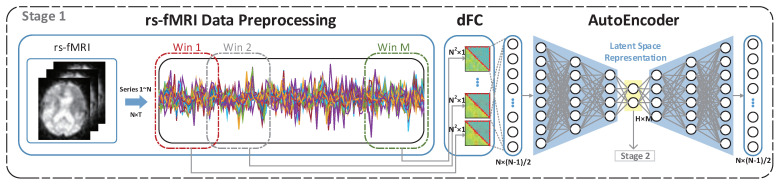
The main processes of the first stage of LSRNet training: (1) rs-fMRI pre-processing, (2) constructing the dFC, and (3) using a deep autoencoder to learn the dFC in order to obtain the latent space representation. In brief, we used the autoencoder to learn and extract the dFC high-level latent space representation.

**Figure 2 brainsci-12-01348-f002:**
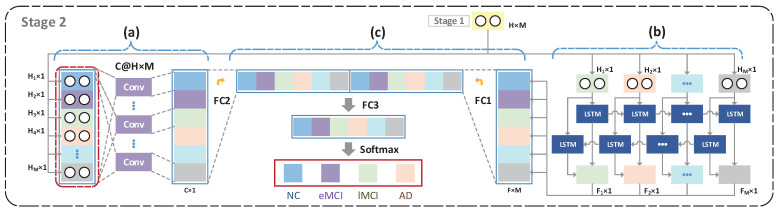
The main processes of the second stage of LSRNet training: (**a**) convolutional neural network for the global perspective, (**b**) BiLSTM network for the local perspective, (**c**) multi-perspective feature fusion network.

**Figure 3 brainsci-12-01348-f003:**
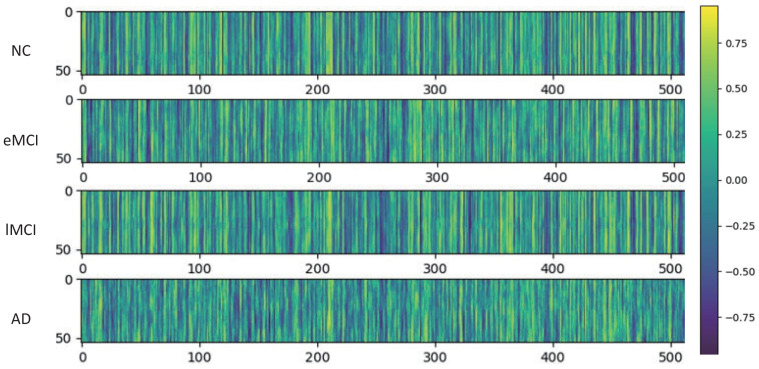
The latent space representation obtained by training the four classification models. The latent space representation of the dFC extracted by autoencoder. The color indicates the value of the feature.

**Table 1 brainsci-12-01348-t001:** Performance of the proposed method on partial metrics (NC vs. eMCT, NC vs. AD) on binary classification.

Method	NC vs. eMCI (%)			NC vs. AD (%)		
	ACC	${\mathrm{ACC}}_{\mathrm{NC}}$	${\mathrm{ACC}}_{\mathrm{eMCI}}$	ACC	${\mathrm{ACC}}_{\mathrm{NC}}$	${\mathrm{ACC}}_{\mathrm{AD}}$
ROIs + SVM	61.2	61.5	60.9	61.7	63.2	55.6
sFC + SVM	61.8	65.4	58.6	64.6	61.8	71.4
dFC + BiLSTM	63.6	63.9	63.2	65.6	63.2	69.2
dFC + GCNN	69.5	58.6	80.0	76.9	73.3	81.8
LSRNet-G	79.7	82.8	76.7	84.6	86.7	81.8
LSRNet-B	84.1	76.2	91.3	88.9	88.2	89.3
LSRNet	84.6	83.8	86.7	95.1	96.7	90.9

**Table 2 brainsci-12-01348-t002:** Performance of the proposed method on partial metrics (eMCI vs. lMCI, lMCI vs. AD) on binary classification.

Method	eMCI vs. lMCI (%)			lMCI vs. AD (%)		
	ACC	ACC${}_{\mathrm{eMCI}}$	ACC${}_{\mathrm{lMCI}}$	ACC	ACC${}_{\mathrm{lMCI}}$	${\mathrm{ACC}}_{\mathrm{AD}}$
ROIs + SVM	58.1	55.6	38.5	55.3	68.0	30.8
sFC + SVM	60.9	75.0	45.5	59.6	68.6	41.2
dFC + BiLSTM	63.0	65.4	60.7	57.1	55.6	62.5
dFC + GCNN	72.0	78.1	61.1	66.6	85.7	35.2
LSRNet-G	74.0	68.8	83.3	75.6	85.7	58.8
LSRNet-B	80.0	90.6	61.1	76.3	68.4	84.2
LSRNet	80.6	83.3	71.4	84.2	87.0	80.0

**Table 3 brainsci-12-01348-t003:** Performance of all methods on four classifications.

Method	NC vs. eMCI vs. lMCI vs. AD (%)				
	ACC	${\mathrm{ACC}}_{\mathrm{NC}}$	${\mathrm{ACC}}_{\mathrm{eMCI}}$	${\mathrm{ACC}}_{\mathrm{lMCI}}$	${\mathrm{ACC}}_{\mathrm{AD}}$
ROIs + SVM	26.4	36.0	28.6	30.8	5.3
sFC + SVM	34.5	47.6	36.8	26.5	30.8
dFC + BiLSTM	46.1	47.6	71.4	10.5	50.0
dFC + GCNN	47.2	66.6	52.4	31.6	39.3
LSRNet-G	49.4	66.7	33.3	21.1	67.8
LSRNet-B	52.8	61.9	57.1	36.8	53.4
LSRNet	57.3	76.2	42.9	52.6	57.1

**Table 4 brainsci-12-01348-t004:** Comparison with other methods.

Target	Authors	Method	Modality	Dataset	Accuracy (%)
NC/eMCI	Jie et al. [21]	Wck-CNN	rs-fMRI	48 NC, 50 eMCI	**84.6**
	Lin et al. [27]	LSTM	rs-fMRI	48 NC, 50 eMCI	84.5
	Our	LSRNet	rs-fMRI	48 NC, 50 eMCI	**84.6**
NC/AD	Jie et al. [21]	Wck-CNN	rs-fMRI	48 NC, 31 AD	88.0
	Wang et al. [13]	STNet	rs-fMRI	48 NC, 31 AD	90.3
	Lin et al. [27]	LSTM	rs-fMRI	48 NC, 31 AD	92.8
	Bi et al. [28]	PCC + SVM	rs-fMRI	36 NC, 25AD	94.4
	Our	LSRNet	rs-fMRI	48 NC, 31 AD	**95.1**

## Data Availability

The subject data used in this study were obtained from the Alzheimer’s Disease Neuroimaging Initiative (ADNI) database (adni.loni.usc.edu).

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
