# Peer review of "Multi-Perspective Feature Extraction and Fusion Based on Deep Latent Space for Diagnosis of Alzheimer’s Diseases"

_brainsci, 2022, doi:10.3390/brainsci12101348_

Round 1

Author Response

Response to Reviewer 1 Comments

Re: Response to reviewers

    We sincerely thank you for your letter and for the reviewers' comments concerning our manuscript entitled "Multi-Perspective Feature Extraction and Fusion Based on Deep Latent Space for Early Diagnosis of Brain Diseases". Those comments are all valuable and very helpful for revising and improving our paper, as well as the important guiding significance to our researches.

    The paper has been linguistically retouched by a native English-speaking editor. We have studied comments carefully and have made correlation which we hope meet with approval. 

Reviewer 2 Report

The paper is well written and discusses Multi-Perspective Feature Extraction and Fusion Based on DeepLatent Space for Early Diagnosis of Brain Diseases the authors used 174 subjects with rs-fMRI from the ADNI dataset, including 48 NC, 50 patients with early MCI (eMCI), 45 patients with late MCI (late MCI, lMCI) and 31 patients with AD. They also used 174 subjects with rs-fMRI from the ADNI dataset, including 48 NC, 50 patients with early MCI (eMCI), 45 patients with late MCI (late MCI, lMCI) and 31 patients with AD. there are 4 tables and 1 figure, the paper's data is available throught the ADNI consortium.  The authors added mathematical equations, and calculated disease predict = FusionNet (concat(GF, DF))

1. Consider splitting figure 1 to fig 1 and 2 (its a little bit overcrowded) and update the legends and text reference accordingly 

2. Add citations to the following papers- 

titleAn EEG Finger-Print of fMRI deep regional activation. authors 

Meir-Hasson Y, Kinreich S, Podlipsky I, Hendler T, Intrator N. PMID: 24246494 journal Neuroimage. 2014 Nov 15;102 Pt 1:128-41. doi: 10.1016/j.neuroimage.2013.11.004. Epub 2013 Nov 15.

3. add abbreviations (eg MCI, PCC etc)

4. Give reference to the ADNI ethics approval for the AD patient study participant if possible 

Author Response

Response to Reviewer 2 Comments

Re: Response to reviewers

We sincerely thank you for your letter and for the reviewers' comments concerning our manuscript entitled "Multi-Perspective Feature Extraction and Fusion Based on Deep Latent Space for Early Diagnosis of Brain Diseases". Those comments are all valuable and very helpful for revising and improving our paper, as well as the important guiding significance to our researches.

The paper has been linguistically retouched by a native English-speaking editor. We have studied comments carefully and have made correlation which we hope meet with approval. The responds to the reviewer's comments are as followings:

Point 1: Consider splitting figure 1 to fig 1 and 2 (its a little bit overcrowded) and update the legends and text reference accordingly 

Response 1: Thank you for your valuable suggestions. Based on your suggestion, we have split Figure 1 into Figure 1 and Figure 2 and updated the legend and text references accordingly. 

Point 2: Add citations to the following papers-title An EEG Finger-Print of fMRI deep regional activation. Authors Meir-Hasson Y, Kinreich S, Podlipsky I, Hendler T, Intrator N. PMID: 24246494 journal Neuroimage. 2014 Nov 15;102 Pt 1:128-41. doi: 10.1016/j.neuroimage.2013.11.004. Epub 2013 Nov 15.

Response 2: Thank you for your valuable suggestions. We have added this paper in the introduction section.

Point 3: add abbreviations (eg MCI, PCC etc)

Response 3: Thank you for your advice. We carefully reviewed and added abbreviations to the text. 

Point 4: Give reference to the ADNI ethics approval for the AD patient study participant if possible 

Response 4: Thank you for your advice. We have added a reference to the ADNI protocol. Informed consent was obtained from all subjects participating in the study in accordance with the ADNI protocol, which can be found in more detail at adni.loni.usc.edu.

Reviewer 3 Report

Major points

1.      Introduction. Lines 20 – 22. Please rephrase to ‚Alzheimer’s Disease (AD) is a chronic neurodegenerative disease that clinically manifests with cognitive and behavioural impairment with gradual decline in activities of daily living‘.

2.      Introduction. Lines 28 – 29. There are no treatments ‚limit the progression of the disease‘. Rephrase the sentence.  

3.      Materials. Line 87 - 88. ‚so that these 174 subjects had a total of 563 scans, which can be classified as 154 NC, 165 eMCI, 145 lMCI and 99 AD‘. That statement is wrong. If the subject has 3 scanning sessions it does not qualify as 3 subjects. It is 1 subject with 3 scanns.

4.      Materials. Please provide the name and type of scanner.

5.      Materials. Please provide other acquisition parameters such as matrix size, FoV.

6.      Materials. Please provide the acquisition parameters of structural T1 images

7.      Methods. Figure 1. Is it LSRNET or LSRNet?

Minor points

1.      Abstract. Line 3. Please capitalize ‚pearson‘

2.      Abstract. Line 16. ‚the results validated the effectiveness of the method‘ is not a proper conclusion. Please revise.

3.      Introduction. Line 25. Alzheimer’s Disease should be changed to AD.

4.      Introduction. Line 29. Alzheimer’s Disease should be changed to AD.

5.      Introduction. Line 30. Change ‚irretrievable‘ to ‚incurable‘

6.      Introduction. Line 34 - 35. Mild Cognitive Impairment and Alzheimer’s disease have already been abbreviated. The terms should be abreviated once, at first mentioning, and then used only the abbreviations throughout the entire manuscript.

7.      Introduction. Line 68. Capitalize ‚second‘.

8.      Introduction. Line 85 - 86. Revise ‚deeply explore the deep contextual information‘

9.      Introduction. Line 74 – 75. This sentence ‚For global contextual relevance features, we propose a convolutional neural network based on a global perspective‘ is similar as the preceding one.

10.  Introduction. Line 76. Define the BiLSTM.

11.  Introduction. Line 80. Define the ADNI dataset.

12.  Materials. Line 84. Correct is ‚we included rs-fMRI data of 174 subjects from the ADNI dataset,‘.

13.  Materials. Line 85. Define what is NC.

Author Response

Response to Reviewer 3 Comments

Re: Response to reviewers

We sincerely thank you for your letter and for the reviewers' comments concerning our manuscript entitled "Multi-Perspective Feature Extraction and Fusion Based on Deep Latent Space for Early Diagnosis of Brain Diseases". Those comments are all valuable and very helpful for revising and improving our paper, as well as the important guiding significance to our researches.

The paper has been linguistically retouched by a native English-speaking editor. We have studied comments carefully and have made correlation which we hope meet with approval. The responds to the reviewer's comments are as followings:

Major points

Point 1: Introduction. Lines 20 – 22. Please rephrase to ‚Alzheimer’s Disease (AD) is a chronic neurodegenerative disease that clinically manifests with cognitive and behavioural impairment with gradual decline in activities of daily living‘.

Response 1: Thank you for your valuable suggestions. The author has rewritten the sentence.

Point 2: Introduction. Lines 28 – 29. There are no treatments ‚limit the progression of the disease‘. Rephrase the sentence.  

Response 2: Thank you for your valuable suggestions. The author has rewritten the sentence.

Point 3: Materials. Line 87 - 88. ‚so that these 174 subjects had a total of 563 scans, which can be classified as 154 NC, 165 eMCI, 145 lMCI and 99 AD‘. That statement is wrong. If the subject has 3 scanning sessions it does not qualify as 3 subjects. It is 1 subject with 3 scanns.

Response 3: Thank you for your valuable suggestions. The author has rewritten the sentence. The correct words should be “These 563 scans could be divided into 154 NC cases, 165 eMCI cases, 145 lMCI cases and 99 AD cases”.

Point 4: Materials. Please provide the name and type of scanner.

Response 4: Thank you for the advice. We have added the name and type of scanner. Each RS-fMRI scan was acquired at a different medical centers using a 3.0T Philips scanner.

Point 5: Materials. Please provide other acquisition parameters such as matrix size, FoV.

Response 5: Thank you for the advice. We have supplemented the paper with other acquisition parameters, such as matrix size, FoV, etc.

Point 6: Materials. Please provide the acquisition parameters of structural T1 images

Response 6: Thank you for the advice. We have supplemented the structured T1 images with acquisition parameters such as matrix size, FoV, etc.

Point 7: Methods. Figure 1. Is it LSRNET or LSRNet?

Response 7: Thank you for your valuable suggestions. The correct name is LSRNet and the author has corrected the error.

Minor points

Point 1: Abstract. Line 3. Please capitalize ‚pearson‘

Response 1: Thank you for the advice. The author has corrected this lapse.

Point 2: Abstract. Line 16. ‚the results validated the effectiveness of the method‘ is not a proper conclusion. Please revise.

Response 2: Thank you for the advice. The author has corrected this lapse.

Point 3: Introduction. Line 25. Alzheimer’s Disease should be changed to AD.

Response 3: Thank you for the advice. The author has corrected this lapse.

Point 4: Introduction. Line 29. Alzheimer’s Disease should be changed to AD.

Response 4: Thank you for the advice. The author has corrected this lapse.

Point 5: Introduction. Line 30. Change ‚irretrievable‘ to ‚incurable‘

Response 5: Thank you for the advice. The author has corrected this lapse.

Point 6: Introduction. Line 34 - 35. Mild Cognitive Impairment and Alzheimer’s disease have already been abbreviated. The terms should be abreviated once, at first mentioning, and then used only the abbreviations throughout the entire manuscript.

Response 6: Thank you for the advice. The author has rechecked the paper and corrected the lapse.

Point 7: Introduction. Line 68. Capitalize ‚second‘.

Response 7: Thank you for the advice. The author has corrected this lapse.

Point 8: Introduction. Line 85 - 86. Revise ‚deeply explore the deep contextual information‘

Response 8: Thank you for the advice. The author has corrected this lapse.

Point 9: Introduction. Line 74 – 75. This sentence ‚For global contextual relevance features, we propose a convolutional neural network based on a global perspective‘ is similar as the preceding one.

Response 9: Thank you for the advice. The author has rewritten the sentence.

Point 10: Introduction. Line 76. Define the BiLSTM.

Response 10: Thank you for the advice. The author has added a definition of the term.

Point 11: Introduction. Line 80. Define the ADNI dataset.

Response 11: Thank you for the suggestion. Since the dataset is defined later, the author has reworked the sentence.

Point 12: Materials. Line 84. Correct is ‚we included rs-fMRI data of 174 subjects from the ADNI dataset.

Response 12: Thank you for the advice. The author has corrected this lapse.

Point 13: Materials. Line 85. Define what is NC.

Response 13: Thank you for the advice. The author has added a definition of the term.

Round 2

Reviewer 2 Report

The paper can now be accepted for publication

Author Response

Re: Response to reviewers

We sincerely thank you for your letter and for the reviewers' comments concerning our manuscript. Those comments are all valuable and very helpful for revising and improving our paper, as well as the important guiding significance to our researches. The paper has been linguistically retouched by a native English-speaking editor. We have studied comments carefully and have made correlation which we hope meet with approval.

The article has been removed from previous revision traces and new ones have been added.

Reviewer 3 Report

The authors replied to raised concerns.

Author Response

(The authors gave the same response as above.)
